**communications** engineering

# A three-electrode dual-power-supply electrochemical pumping system for fast and energy efficient lithium extraction and recovery from solutions

Kazuya Sasaki [1,2] ✉, Kiyoto Shin-mura[1,2], Shunsuke Honda[1], Hirofumi Tazoe[3] & Eiki Niwa [1]

The demand for Li-ion batteries (LIBs) for use in electric vehicles, which is key to realizing a decarbonized society, is accelerating. However, the supply of Li resources has recently become a major issue, thereby necessitating the development of economical and sustainable technologies of brine/seawater-based Li extraction and recycling Li from spent LIBs. This paper presents an innovative electrochemical pumping technology based on a new cell structure for Li extraction/recovery. This system can provide large electrochemical driving forces while preventing the occurrence of electronic conduction due to electrolyte reduction. This electrochemical pumping system allows extraction/recovery of Li ions from the anode side to the cathode side, rather than the diffusion of other ions, due to the ion-diffusion-bottleneck size of the electrolyte material. Using this system, high-purity Li can be collected with high energy efficiency and at least 464 times faster than that via conventional electrochemical pumping, even with a commercially available Li-ion electrolyte plate.

The use of rechargeable batteries has accelerated in the 21st century. In particular, the market share of Li-ion batteries (LIBs) is significantly expanding owing to their high performance, characterized by their high energy and power densities and long life[1–5]. Furthermore, Li resources are anticipated to be used as fuel for thermonuclear fusion power generation after 2035[6–9]. Until 2017, most Li resources were extracted from brine via long-term evaporation. However, the majority of the recent supply is derived from mines for temporarily satisfying the rapidly increasing demand. The supply from mines can tackle the rapid increase in demand owing to the short construction period of this approach. However, problems such as a large environmental impact and high extraction and refining costs are encountered in this strategy. Thus, to satisfy the rapidly growing demand for Li resources, the development of innovative Li extraction technologies is crucial for enabling an economical, safe, and sustainable supply of Li. The European Union is considering mandating the extraction, recovery, and recycling of Li resources as battery materials[10]. However, the recovery of battery-grade high-purity Li using conventional refining methods is

difficult. Therefore, technologies that can permit the recovery of Li from spent LIBs must be urgently developed. New Li-extraction/recovery technologies, such as adsorption/dissociation (ion sieving)[11–22], ion exchange[23–28], precipitation[29–33], and liquid–liquid extraction[34–40], have been proposed in this regard. However, these technologies are unsuitable for industrial processes because of their small extraction volumes, low recoveries, and inferior element selectivity.

Electrochemical pumping using Li-ion conductors shows promise as an alternative Li-extraction/recovery technology[41–48]. In this method of resource extraction/recovery, the desired ions in a tank—which are fractionated by a diaphragm with ionic conductivity—are transferred and extracted to another tank owing to a difference in electrochemical potential. Li-containing solutions, such as aqueous solutions of the Li ions from spent LIBs dissolved in acid, brine, or seawater, can be used in this regard. Densely sintered Li-ion conductors, such as lithium lanthanum titanates (LLTO; $La_{2/3-x}Li_{3x}TiO_3$) and ionic liquids, are typically used as diaphragms. When a sufficiently large potential difference for the electrolysis of water is

[1]Graduate School of Science and Technology, Hirosaki University, 3 Bunkyo-cho, Hirosaki, Aomori 036-8561, Japan. [2]Lithium Resources Research Organization, Hirosaki University, 3 Bunkyo-cho, Hirosaki, Aomori 036-8561, Japan. [3]Institute of Radiation Emergency Medicine, Hirosaki University, 61-1 Honmachi, Hirosaki, Aomori 036-8564, Japan. ✉e-mail: k_sasaki@hirosaki-u.ac.jp

applied between the electrodes sandwiching the diaphragm, Li ions migrate from the anode side to the cathode side via oxygen and hydrogen gas-generating reactions at the anode and cathode, respectively.

Li extraction/recovery via electrochemical pumping offers several advantages over other procedures, with the main benefit being its high Li-ion selectivity. The adsorption and coprecipitation methods are inferior in this regard as they lead to contamination with alkali metal ions, such as Na and K, which have similar chemical properties to those of Li ions. The ion selectivity of electrochemical pumping using $La_{0.55}Li_{0.35}TiO_3$ as a diaphragm was evaluated by Kunugi et al.[46]. In their study, a mixed aqueous solution of lithium chloride, sodium chloride, potassium chloride (0.1 mol/L) was placed in the anode-side tank, whereas distilled water was placed in the cathode-side tank. Inductively-coupled-plasma atomic emission spectrometry (ICP-AES) analysis of the cations transferred to the cathode side during 100 h electrochemical pumping revealed that Na and K ions did not migrate. This suggests that electrochemical-pumping-based Li extraction/recovery performed using an LLTO diaphragm with $x \approx 0.1$ achieves high Li-ion selectivity. The reason why only Li diffuses in LLTO is due to the crystal structure of LLTO. LLTO has a perovskite-type structure and the La-rich and -poor layers are stacked in the c-axis direction[49,50], and Li ions are located at the La sites of the La-poor layer. For a Li ion to migrate to the neighboring La site, it must pass through a plane surrounded by oxide ions, which represents a bottleneck for Li-ion diffusion. The size of this LLTO bottleneck is larger than that of a Li ion and smaller than that of other alkali ions such as Na and K. Therefore, the electrochemical pumping system can only transfer Li ions from the anode tank to the cathode tank. Another advantage of electrochemical pumping is its high collection rate, which can be achieved in principle by increasing the electrochemical potential difference, that is, the driving force for ion migration. Unlike other methods that require large devices, electrochemical pumping can achieve high-speed collection using relatively small instruments. The third advantage of this process is its excellent economy. Li extraction/recovery from aqueous basic solutions via electrochemical pumping produces oxygen and hydrogen gases through the reactions expressed in Eqs. (1) and (2) at the anode and cathode, respectively. Under ideal operating conditions, the energy efficiency of Li collection is extremely high because aqueous solutions of high-purity lithium hydroxide (LiOH) can be simply obtained by consuming the energy required for generating these gases.

$$4Li^+(solution) + 4OH^- \rightarrow 4Li^+(solid) + 4e^- + 2H_2O + O_2 \uparrow \quad (1)$$

$$2Li^+(solid) + 2e^- + 2H_2O \rightarrow 2Li^+(solution) + 2OH^- + H_2 \uparrow \quad (2)$$

Furthermore, electrochemical pumping systems have a simple structure, which permit straightforward and cost-effective construction of large-scale plants. This system can continuously recover lithium without the replacement of absorbent materials or solutions or switching between charging and discharging cycles as in other lithium recovery methods, significantly reducing operational costs. Moreover, the generated high-purity hydrogen and oxygen gas can be recovered and used as fuel.

However, the energy efficiency of electrochemical pumping through LLTO electrolytes in a conventionally structured cell abruptly decreases when a large potential difference over a certain threshold is applied to increase the Li collection rate. High applied voltages cause the tetravalent-to-trivalent reduction of Ti near the cathode-side surface of the LLTO electrolyte. Consequently, electronic conduction occurs in the LLTO electrolyte and the energy efficiency is significantly reduced owing to heat generation (electronic conduction-induced Joule heating), which is unrelated to the amount of collected Li. Therefore, to encourage the development of industrial-scale systems for electrochemical pumping-based Li extraction/recovery, the reduction of the constituent ions of the electrolyte must be suppressed even when a large voltage is applied to achieve a high collection rate.

To this end, a new electrochemical pumping system that can increase the Li extraction/recovery rate while maintaining a high energy efficiency was designed in the present study (Fig. 1a). Conventional electrochemical pumping systems comprise a pair of electrodes placed across an electrolyte diaphragm and a single power supply that provides a potential difference to the electrodes[41–48]. In the new system, a third electrode was placed in the cathode-side tank, and the desired DC voltage was generated and applied using a secondary power supply situated between the electrode on the cathode-side surface of the diaphragm and the third electrode.

A simplified equivalent circuit for the new electrochemical pumping system (Fig. 1a) is shown in Fig. 1b. In principle, this system can collect Li at a limitlessly high rate via three mechanisms, which are explained below using equivalent circuits. The first mechanism involves an increase in the dissolution rate of Li ions in the LLTO electrolyte due to the increased overpotential of slow process reactions involving the oxygen or hydrogen gas-generating reactions caused by the subcircuit construction. In electrochemical reactions, the reaction rate increases with reaction overpotential. As described later, increasing the voltage of the secondary power supply, or decreasing the solution impedance of the newly added subcircuits increased the overpotential of the gas-generating reactions and enhanced the Li-ion transfer rate. The second mechanism involves an increase in the electrochemical reaction rate owing to an increase in the reaction field for hydrogen gas generation, which is a relatively slow process. In a conventional electrochemical pumping cell, the reaction field for hydrogen gas generation is the surface of the electrode prepared on the LLTO electrolyte, with the area of the two-dimensional diaphragm limiting the extent of the field. In contrast, in the new electrochemical pumping cell (Fig. 1a), the reaction field for hydrogen gas generation is the entire surface of the mesh or sponge-like electrode that spreads three-dimensionally to a certain distance from the LLTO surface. In other words, the reaction field for the slow hydrogen gas-generation process can be effortlessly and remarkably increased, thereby enhancing the electrochemical reaction rate. The third mechanism involves maintaining a voltage balance between the main and secondary power supplies, which prevents the electronic conduction in the LLTO electrolyte even when the main power-supply voltage is limitlessly increased. Applying a sufficiently large voltage from the secondary power supply to the new system results in $I_3 \geq I_1$ (Fig. 1b). When $I_3 = I_1$, only the reaction expressed in Eq. (3) occurs at the second electrode without gas generation. When $I_3 > I_1$, the reaction described in Eq. (4) occurs at the electrode on the cathode-side surface of the LLTO connected to the positive side of the secondary power supply.

$$4Li^+(solid) \rightarrow 4Li^+(solution) \quad (3)$$

$$4Li^+(solid) + 4OH^- \rightarrow 4Li^+(solution) + 4e^- + 2H_2O + O_2 \uparrow \quad (4)$$

Generally, the potential of the cathode-side surface of LLTO does not drop below the hydrogen gas-generating potential (0 V vs. reversible hydrogen electrode (RHE)) and is always higher than the tetravalent-to-trivalent reduction potential of Ti (0.0 V vs. RHE). Therefore, the main power-supply voltage applied to the LLTO electrolyte can be arbitrarily increased while suppressing the electronic conduction by appropriately controlling the voltage balance between the main and secondary power supplies. The increase in the main power-supply voltage enhances the electrochemical reaction rate by significantly increasing the overpotential of the oxygen gas-generating reaction at the anode, which is relatively slow. The Joule heating losses due to electronic conduction were not significant, and the energy efficiency did not experience an abrupt drop.

Based on the obtained experimental results, the effects of the mechanisms of high-speed high-energy-efficiency Li collection were elucidated and a new electrochemical pumping system exhibiting a limitlessly increasing Li collection rate was devised.

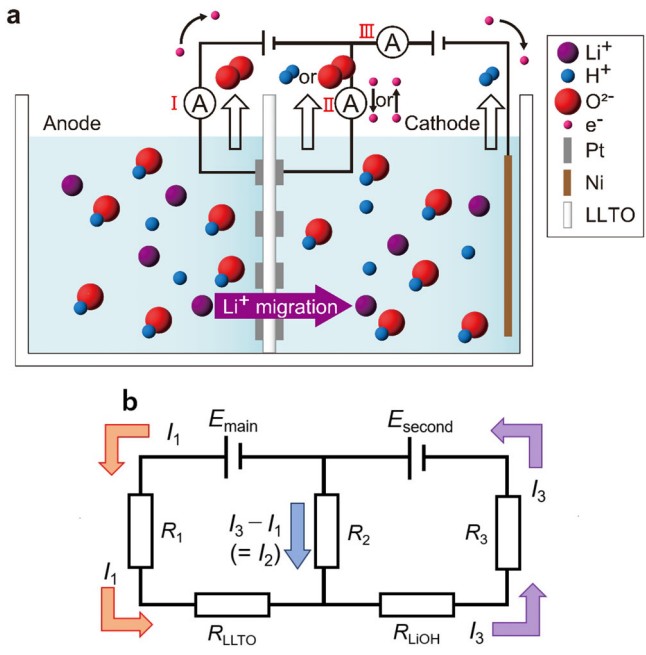

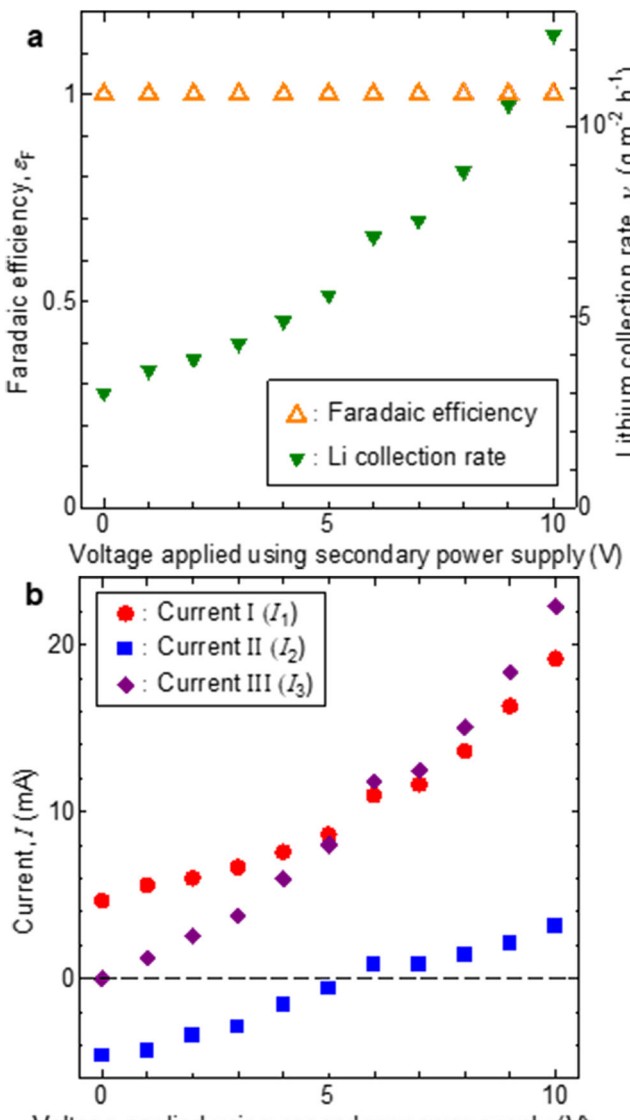

**Fig. 1 | The proposed new system of electrochemical pumping. a** Schematic of newly designed electrochemical pumping cell comprising a La$_{0.57}$Li$_{0.29}$TiO$_3$ (LLTO) electrolyte, Pt anode and cathode, a third Ni electrode, and main and secondary power supplies, thereby enabling mass transfer during Li ion extraction/recovery. The mentioned voltages represent those determined experimentally and not the theoretical limits. **b** Equivalent circuit of the new electrochemical pumping system: $E_{main}$, main power supply; $E_{second}$, secondary power supply; $R_1$, impedance of anodic reaction on the first electrode; $R_2$, impedance of anodic or cathodic reaction on the second electrode; $R_3$, impedance of cathodic reaction on the third electrode; $R_{LLTO}$, impedance of the LLTO electrolyte; $R_{LiOH}$, impedance of aqueous LiOH solution; $I_1$ and $I_3$, currents.

## Results
### Effects of additional subcircuit

Figure 2a shows the dependence of the Li collection rate and faradaic efficiency on the secondary power-supply voltage in the new electrochemical pumping system. In our previous study[45], which was also conducted using the LLTO electrolyte diaphragm and Pt electrodes used in the present study, no electronic conduction occurred in the LLTO electrolyte, and the Li-ion transference number was 1 when the main power-supply voltage was below 2.0 V. The distance between the second and third electrodes and the concentration of the cathode-side LiOH solution were 57 mm and 0.001 mol/L, respectively. The values for a conventional electrochemical pumping system without the secondary power supply and third electrode were obtained using a secondary power-supply voltage of 0 V. When the main power-supply voltage was 2.0 V, the faradaic efficiency remained constant regardless of the secondary power-supply voltage, and the Li collection rate increased with increasing secondary power-supply voltage. Increasing the secondary power-supply voltage led to hydrogen gas generation owing to the remarkably rapid reduction of protons at the third electrode, which was confirmed by naked-eye observation, although this was not evident on the surface of the cathode in the conventional system (secondary power-supply voltage = 0 V).

The increase in the Li recovery rate can be explained using the subcircuit constituent factors $E_{second}$, $R_2$, $R_{LiOH}$, and $R_3$ in the simplified equivalent circuit (Fig. 1b). Equations (5) and (6) describe the relationship between the power supply voltages, impedance of the elementary reaction process, and currents.

$$E_{main} = R_1 I_1 + R_{LLTO} I_1 - R_2(I_3 - I_1)$$
$$= (R_1 + R_{LLTO} + R_2)I_1 - R_2 I_3, \quad (5)$$

**Fig. 2 | Dependence of characteristics of new electrochemical pumping system on secondary power-supply voltage. a** faradaic efficiency, Li collection rate, and **b** currents monitored by ammeters I, II, and III (shown in Fig. 1) (main power supply voltage, 2.0 V; anode-side solution, 1.0 mol/L aqueous LiOH; cathode-side solution, $1 \times 10^{-3}$ mol/L aqueous LiOH; distance between the second and third electrodes, 57 mm).

$$E_{main} + E_{second} = R_1 I_1 + R_{LLTO} I_1 + R_{LiOH} I_3 + R_3 I_3$$
$$= (R_1 + R_{LLTO})I_1 + (R_{LiOH} + R_3)I_3, \quad (6)$$

where $E_{main}$, $E_{second}$, $R_1$, $R_{LLTO}$, $R_2$, $R_{LiOH}$, $R_3$, $I_1$, $I_3 - I_1$, and $I_3$ are the main power-supply voltage; secondary power-supply voltage; impedance of the anodic reaction on the first electrode; impedance of Li-ion conduction via LLTO; impedance of the reaction on the second electrode, which was caused by the cathodic (H$_2$ generation and elution of Li$^+$ into the solution) or anodic reactions (O$_2$ generation and elution of Li$^+$ into the solution); resistance of the solution in the cathode-side tank; impedance of the hydrogen gas-generating reaction at the third electrode; and currents monitored using ammeters I, II, and III, respectively. $R_1 I_1$, $R_{LLTO} I_1$, $R_2(I_3 - I_1)$, $R_{LiOH} I_3$, and $R_3 I_3$ represent the overpotential of the anodic reaction, difference between the Fermi potentials of the two electrolyte surfaces (electrolyte overpotential), overpotential of the cathodic reaction, potential difference between the second and third electrodes in the LiOH

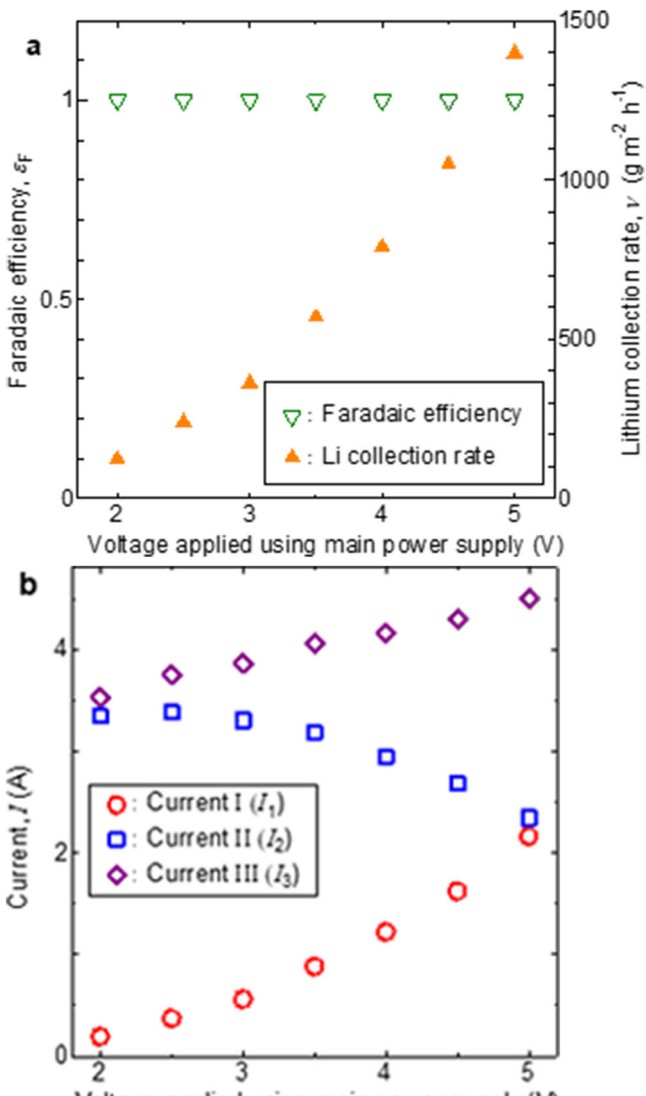

**Fig. 3 | Dependence of characteristics of new electrochemical pumping system on main power-supply voltage.** Dependence of **a** faradaic efficiency, Li collection rate, and **b** currents determined using ammeters I, II, and III (Fig. 1) (secondary power supply voltage, 10.0 V; anode-side solution, 1.0 mol/L aqueous LiOH; cathode-side solution, 1.0 mol/L aqueous LiOH; distance between the second and third electrodes, 57 mm).

solution in the cathode-side tank, and overpotential of the hydrogen gas-generating reaction at the third electrode, respectively. The direction of the current, as depicted in Fig. 1b, was opposite to the direction of the electron flow, as shown in Fig. 1a. All impedances were considered to be constant and independent of the voltage. Because $R$ and all $I$ had positive values, $I_1$ and $I_3$ increased with increasing $E_{second}$, as indicated by the third term in Eq. (6). Because $E_{main}$ was constant in the experiments that yielded the results shown in Fig. 2, $I_1$ also increased with increasing $I_3$, as indicated in Eq. (5). In other words, both $I_1$ and $I_3$ increased with increasing $E_{second}$. The Li-ion transference number was 1 under the experimental conditions that led to the results shown in Fig. 2; thus, according to Faraday's law, $I_1$ was equivalent to the Li collection rate. Therefore, the increase in the Li collection rate (Fig. 2a) was due to the construction of the subcircuit (Mechanism I) comprising a secondary power supply and additional electrode. Phenomenologically, this was caused by two factors: an increase in the anodic reaction rate caused by the overpotential of the anodic reaction ($R_1 I_1$) at the first electrode, and an increase in the cathodic reaction rate caused by the overpotential of the

cathodic reaction ($R_3 I_3$) at the third electrode, which result from the increase in $I_1$. This effect was clarified by analyzing the dependence of the Li collection rate on the distance between the second and third electrodes and the concentration of the cathode-side LiOH solution (Supplementary Fig. 1a and b, respectively). In both experiments, the main and secondary power-supply voltages were 2.0 and 5.0 V, respectively. A cathode-side LiOH solution concentration of 1.0 mol/L and a distance between the second and third electrodes of 57 mm were used to obtain the data shown in Supplementary Fig. 1a and 1b, respectively. The Li collection rate monotonically increased with decreasing distance between the second and third electrodes (Supplementary Fig. 1a) and increasing concentration of the cathode-side LiOH solution (Supplementary Fig. 1b). The decreasing distance between the electrodes as well as the increasing conductive carrier density due to the increase in LiOH solution concentration diminished the resistance of the cathode-side solution between the second and third electrodes ($R_{LiOH}$). Consequently, $I_1$ and $I_3$ increased even when $E_{main}$ and $E_{second}$ were fixed, as indicated by the second term in Eq. (6). These phenomena can be explained by the fact that the aforementioned increases in current were due to enhanced rates of elementary reactions and ion diffusion, which were caused by the increase in the overpotential of the elementary reactions owing to the reduction in the potential difference between the second and third electrodes. The results shown in Supplementary Fig. 1b indicate that the Li collection rate in the new system did not decrease significantly, even if the chemical potential difference decreased owing to the increase in Li-ion concentration of the cathode-side solution as the Li-ion migration proceeded. This feature is another advantage of the new system, considering that the Li concentration of the cathode-side solution must be high to permit industrial production of lithium carbonate or LiOH powders.

**Limitations of new system in terms of Li collection rate**

In this section, the dependence of $I_1$, $I_2$, and $I_3$ on the secondary power-supply voltage (Fig. 2b) is examined and the limitations involved in balancing the main- and secondary power-supply voltages are discussed. As inferred from the simplified equivalent circuit (Fig. 1b), the sum of $I_1$ and $I_2$ is $I_3$, regardless of the direction of voltage or current. When $E_{main}$ was equal to or higher than the theoretical electrolysis voltage of $H_2O$, $I_1$ and $I_3$ were always positive, and oxygen and hydrogen gases were always generated at the first and third electrodes, respectively, regardless of the power-supply voltage. However, the direction of $I_2$ was reversed by the secondary power-supply voltage. When the main power-supply voltage was fixed at 2.0 V, $I_2$ was always positive if the secondary power-supply voltage was equal to or greater than 6.0 V. That is, a sufficiently high secondary power-supply voltage above a certain threshold ensured that the potential of the second electrode was always higher than the oxygen gas-generating potential, thereby preventing the reduction of the LLTO electrolyte. However, $I_2$ was always negative at secondary power-supply voltages below 5.0 V, and hydrogen gas was generated at the second electrode. Moreover, although the main power-supply voltage was constant, all gas generation speeds varied with the secondary power-supply voltage to such an extent that they could be ascertained visually. Considering that the gas-generation reaction rate varied with the reaction overpotential (that is, the potential of the reaction field), the cathode-side surface potential of the electrolyte diaphragm changed with the secondary power-supply voltage. Overall, these results indicate that a high secondary power-supply voltage increased the upper limit of the main power-supply voltage at which a positive cathode-side surface potential could be maintained for the electrolyte diaphragm.

The changes in faradaic efficiency, Li collection rate (Fig. 3a), and currents (Fig. 3b) when the secondary power-supply voltage was fixed at 10.0 V and the main power-supply voltage was varied were subsequently analyzed. When the main power-supply voltage was increased from 2 to 5 V, the faradaic efficiency remained 1. This means that the applied voltage between the electrodes was successfully enhanced without the occurrence of electronic conduction in LLTO. By increasing the main power-supply

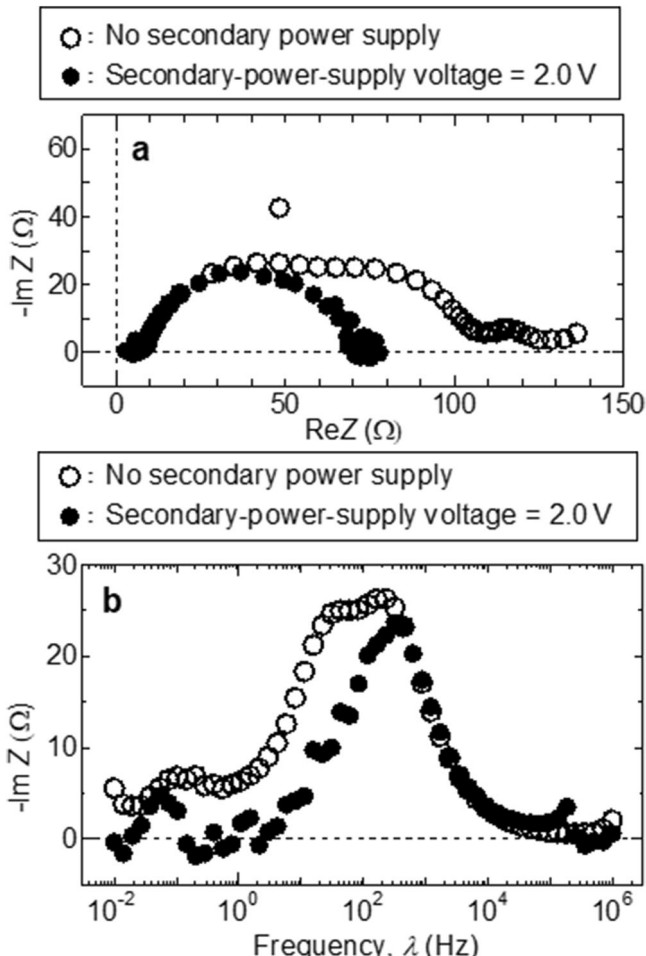

**Fig. 4 | Complex impedance spectra for conventional and new systems (without and with secondary-power-supply voltage of 2.0 V, respectively). a** Nyquist and **b** Bode plots obtained during electrochemical pumping between the first and second electrodes with the LLTO electrolyte. A main power-supply voltage of 2.0 V was used.

voltage from 2 to 5 V, the Li collection rate increased dramatically from 0.12 to 1.40 kg/m²/h, and the energy efficiency increased from 0.20 to 1.44 mmol/Wh. Furthermore, not only the lithium recovery rate but also the energy efficiency increased with increasing main power-supply voltage. The currents shown in Fig. 3b are approximately two orders of magnitude larger than those in Fig. 2b. As shown in Fig. 3b, $I_1$ and $I_3$ were always positive and increased with increasing main power-supply voltage. In contrast, $I_2$ gradually decreased with increasing main power-supply voltage. Notably, even when the main power-supply voltage was 5.0 V, $I_2$ was positive and oxygen gas was generated at the second electrode. The main power-supply voltage at which $I_2$ became negative, which was estimated by extrapolating the plot in Fig. 3b, was approximately 7 V or higher.

Overall, these results suggest that by increasing the secondary power-supply voltage to an appropriate value, the main power-supply voltage can be increased indefinitely while preventing electronic conduction in the electrolyte diaphragm. Essentially, this new electrochemical pumping system can achieve arbitrarily high Li extraction/recovery without the abrupt drop in energy efficiency due to the reduction in faradaic efficiency. In practice, this was confirmed by analyzing the dependence of the faradaic efficiency and Li migration rate on the main power-supply voltage (Fig. 3a). In conventional systems, electronic conduction is known to occur in the LLTO electrolyte at main power-supply voltages above 2.5 V, which lead to an abrupt decrease in the faradaic efficiency[42]. However, in the new system operated at a secondary power-supply voltage as high as 10.0 V, a faradaic

efficiency of 1 was maintained even when the main power-supply voltage was 5.0 V. The Li collection rate in this system was 1.40 kg/m²/h, which was 464 times higher than the maximum value (3.01 g/m²/h) obtained using a conventional system operated at a main power-supply voltage of 2.0 V; the value obtained by establishing contact between the first and second electrodes and the LLTO electrolytic diaphragm is greater than that (2.40 g/m²/h) reported by Zheng et al.[29]. Because these voltages can be further increased by maintaining the balance between the main and secondary power-supply voltages, as described above, the Li collection rate is theoretically not limited to the values reported herein and can be extremely high.

### Enhancement of the reaction field area for rate-limiting processes

Nyquist (Fig. 4a) and Bode plots (Fig. 4b) were acquired by performing two-pole complex impedance measurements between the first and second electrodes using a LiOH solution (1.0 mol/L) in both the anode- and cathode-side tanks. The main power-supply voltage was 2.0 V. The open and closed symbols in Fig. 4 correspond to the conventional (no secondary power supply) and new systems (secondary power supply voltage = 2.0 V), respectively. Three arcs with peak frequencies of approximately $10^{-1}$, $10^{1.5}$, and $10^{2.5}$ Hz were observed in the spectrum of the conventional system. Our previous study[40] revealed that the arc with a peak frequency of approximately $10^{2.5}$ Hz corresponds to the grain-boundary impedance of the LLTO electrolyte, and that the other two arcs in the low-frequency region represent electrode reactions accompanied by gas generation. The arc with a peak frequency of approximately $10^{1.5}$ Hz became negligibly small after the secondary power-supply voltage was applied. This is because the hydrogen gas-generating reaction occurred at the third electrode, which had a large surface area and reaction field. Consequently, the hydrogen gas-generating reaction presumably exhibited an increased rate and no longer dominated the Li collection rate. Therefore, the total impedance was also considerably reduced and the new system containing the third electrode with a large surface area was presumed to dramatically increase the Li collection rate (Mechanism III). Moreover, the intensity of the arc with a peak frequency of $\sim 10^{-1}$ Hz decreased slightly because the reaction field for oxygen gas generation increased, as oxygen gas was generated at the first as well as the second electrodes.

### Discussion

In this study, a novel electrochemical pumping technique was devised for Li extraction/recovery. The cell includes two power supplies, three electrodes, and a La$_{0.57}$Li$_{0.29}$TiO$_3$ (LLTO) electrolyte. The unique structure and behavior of the cell permitted the application of a positive voltage from a third electrode in the catholyte solution via a secondary power source to a conventional cell comprising an electrolyte diaphragm attached to a pair of electrodes. LLTO is an ideal solid electrolyte for the extraction/recovery of high-purity Li that prohibits permeation of cations other than Li ions. Notably, the newly devised system is flexible in terms of accommodating diverse, commercially available electrolytes other than LLTO. The additional subcircuit comprising a secondary power supply, second electrode, and third electrode enhances the Li collection rate of the main circuit, which consists of the main power supply and first and second electrodes. Furthermore, an appropriately high secondary power-supply voltage enables the provision of an unlimitedly high voltage to the main power supply, while ensuring that the potential of the electrolyte diaphragm remains high to prevent reduction. Consequently, this new system theoretically exhibits unlimited potential for rapid Li collection.

In summary, an innovative electrochemical pumping system for Li resource extraction/recovery was designed and comprehensively assessed in terms of its operating conditions. The new system comprising two power supplies, three electrodes, and an LLTO electrolyte enabled Li collection that was at least 464 times faster than that of conventional systems, which was because of the added subcircuits. Additionally, the operating conditions for

achieving high performance were clarified. The Li collection rate was shown to limitlessly increase in principle by controlling the voltage balance between the main and secondary power supplies. Additionally, the operating conditions for high-performance Li recovery were optimized. The development of new configurations and establishment of operating condition constraints for ultrafast Li collection will facilitate the industrial extraction/recovery of Li resources from brine, seawater, or waste LIBs.

## Methods

### Lithium-ion conductor diaphragm (LLTO) and electrodes

A commercially available polycrystalline densely sintered plate of $La_{0.57}Li_{0.29}TiO_3$ (LLTO; $50 \times 50 \times 0.50$ mm³, Toho Titanium Co., Ltd., Japan) was used as the Li ion-conducting electrolyte diaphragm in the electrochemical pumping system. The vendor-published values for relative density (99%) and total ionic conductivity ($5.0 \times 10^{-4}$ S/cm) were used. Grid-shaped high-purity Pt electrodes ( > 99.95 mass%) with the width line and space of each 0.5 mm in a 20 mm × 20 mm were symmetrically placed on either side of the LLTO plate as the first and second electrodes. The electrodes were prepared by screen printing and firing Pt paste TR-7907 (Tanaka Kikinzoku Kogyo K. K., Japan). Pt current collector wires (purity: >99.95 mass%, diameter: 0.50 mm) were attached to the ends of the Pt electrodes using Pt paste TR-7603T (Tanaka Kikinzoku Kogyo K. K., Japan). All Pt pastes were fired in air at 1100 °C for 1 h. To prevent the thermal-stress-induced cracking of the LLTO plate, both heating and cooling were performed at a low rate of 1 °C/min. The procedure employed to prepare the Pt electrodes has been detailed in our previous report[45]. A commercially available Ni mesh (wire diameter: 0.10 mmᵒ; 2.0 × 2.0 cm², Nilaco, Japan) was placed at the desired distance from the LLTO electrolyte and used as the third electrode.

### Setup and evaluation of electrochemical pumping system

The electrochemical pumping cell (Fig. 1a) was fabricated using acrylic plates. A polypropylene spacer that was 0.1 mm thinner than the electrolyte plate was wrapped around the electrolyte plate and then sandwiched with silicone rubber from both sides to completely prevent liquid leakage. During the experiments, the solution was stirred at a constant speed using magnetic stirrers.

A multichannel potentiostat/galvanostat VMP-3 (Bio-Logic, France) was used to apply a potential difference between the electrodes and measure the current. When performing electrochemical pumping with a large current that exceeded the specifications of the VMP-3 instrument, a stabilized power supply TB80V14A360W (Matsusada Precision, Japan) was used to apply voltage, and the voltage and current were accurately measured using a digital multimeter 34461 A (KEYSIGHT, Japan). Under each experimental condition, chronoamperometry was performed until the current value stabilized (maximum measurement time: 1 hour). Under each experimental condition, chronoamperometry was performed until the current value stabilized (maximum measurement time: 1 hour). This measurement was performed once for each experimental condition. Then, under the same applied voltage conditions, impedance spectra were recorded with an alternating voltage of 10 mV in a frequency range from 0.10 Hz to 1.0 MHz.

The Li-ion concentration of the cathode-side solution after 1 h of electrochemical pumping was determined by ICP-AES (SPECTROBLUE® FMX26, HITACHI, Japan, or Optima 7000DV, Perkin Elmer, USA), and the molar mass of transferred Li, $n$, was estimated from the Li-ion concentration of the cathode-side solution. The faradaic efficiency $\varepsilon_F$ was calculated using Eq. (7) based on the time-integrated value of current $I$ during the electrochemical pumping ($\Delta q$ (C)) and $n$ (mol).

$$\varepsilon_F = \frac{nF}{\Delta q} = \frac{nF}{\int I_1 dt} \quad (7)$$

where $F$ is the Faraday constant. A LiOH solution was prepared by dissolving lithium hydroxide powder (LiOH·H₂O, ≥98%; Kanto Chemical) in distilled water. Both sides of the acrylic tanks had identical volumes (130 mL). The average Li-ion collection rate $v$ (g/m²/h) was calculated based on the amount of Li ions transferred from the anode side to the cathode side ($M$ (g)), electrochemical pumping time, and projected electrode area calculated using Eq. (8).

$$v = M/At, \quad (8)$$

where $A$ and $t$ are the projected electrode area (m²) and electrochemical pumping time (h), respectively. The decrease in the Li-ion concentration of the anode-side solution during the electrochemical pumping experiments was assumed to be negligible. The temperature of the solution was maintained at 20 °C.

## Data availability

The authors declare that all experimental data and relevant analysis of this work are available from the corresponding author (HAD) upon reasonable request.

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

## Acknowledgements

This study was supported by JSPS KAKENHI grants (nos. JP19H02639 and JP22H01999), the Takahashi Foundation for Industrial and Economic Research, Hirosaki University Next Generation Institutional Research grants, Dowa Holdings Corporation, Chubu Electric Power Co., Inc., and Toyota Motor Corporation.

## Author contributions

Kazuya Sasaki: conceptualization; funding acquisition; methodology; project administration; supervision; resources; investigation; writing – original draft; writing – review & editing. Kiyoto Shin-mura: methodology; visualization; data curation; investigation; validation. Shunsuke Honda: investigation. Hirofumi Tazoe: investigation (ICP-AES). Eiki Niwa: visualization; writing – review & editing.

## Competing interests

The authors declare no competing interests.
