## [Peer Review File · Communications Engineering]

Reviewers' comments:

Reviewer #1 (Remarks to the Author):

The article titled: "A three-electrode dual-power-supply electrochemical pumping system for ultrafast high-energy-efficiency lithium extraction and recovery" describes the development of a new electrochemical cell set-up allowing faster and improved lithium recovery. The design of the cell is innovative and the research field of great interest; however, additional experiments and clarifications are required before its publication in Communications Engineering.

The authors claim that "the Li collection rate of this new system can be limitlessly increased." I doubt the LLTO solid electrolyte will allow an unlimited rate. The lithium diffusion coefficient in its structure is limited. The lithium must migrate through its structure to reach the cathode side, which will roll the rate. What is the value of the lithium diffusion coefficient in LLTO, and how would it affect the "limitlessly" rate proposed?

The authors also mentioned an "ultrafast high-energy-efficiency." However, I have some doubts about the suitability of such terminology. Increasing the recovery rate by increasing the voltage applied must have a penalty in the energy consumption required to extract the lithium. There is no calculation regarding the energy needed to extract the lithium in the manuscript; such a value will have a tremendous impact on the potential applicability of this new set-up. These calculations must be included, and the energy consumption per mol of lithium must be included.

The manuscript shows the improvement obtained with this new set-up; however, the electrolyte used was pure lithium electrolyte. In real samples, lithium concentration is significantly smaller than other cations present in solution (Na, K, Mg...). The increment in the lithium rate could also affect the LLTO capacity to discriminate between these cations; therefore, the method's selectivity in the conditions described must be included, showing the concentration on the cathode-side of the different cations. If other co-cations are co-transported with the increment of the rate, the capacity of this methodology to extract lithium will be compromised; therefore, such measurements are critical for correctly analyzing the proposed method.

On the other hand, increasing the rate will also increase the accumulation of cations on the LLTO surface since these co-cations will be blocked because of LLTO lithium selectivity. How will such charge accumulation affect the recovery process?

How much is the final lithium concentration reached after applying the electrochemical method?

Finally, there are numerous articles directly related to the manuscript topic that should be included in the manuscript: e., g. Recent reviews about the electrochemical ion pumping technology: <https://doi.org/10.1016/j.coelec.2021.100778>; DOI: 10.1002/adma.201905440; or alternative electrochemical cells for lithium recovery: <https://doi.org/10.1016/j.desal.2019.114192>
Values like the selectivity, purity, energy efficiency, energy consumption, and final Lithium concentration in the present work must be compared with the suggested bibliography to analyze the proposed

methodology's advantages properly.

Reviewer #2 (Remarks to the Author):

This reviewer finds the present manuscript not suitable for publication for the reasons explained below. Major revision of the manuscript is required before publication.

The authors present a new experimental set up with respect to their previous publication in which an LLTO solid lithium conducting membrane with two platinum electrodes deposited in both faces separates two LiOH solutions of different concentration, on the anode side 1 M LiOH and on the cathode side 0.001 M LiOH solution, with an applied voltage 2.0 V across the LLTO membrane.

In the new set up presented in the manuscript, a third Ni cathode and a second power supply are introduced. Upon applying a second voltage with the second power supply, the real impedance (R_{ct}) of the LLTO decreases and the lithium collection rate increases.

According to data in Figure 1 anode-side solution, 1.0 mol/L aqueous LiOH solution; cathode-side solution, 1×10^{-3} mol/L aqueous LiOH solution; distance between the second and third electrodes, 57 mm. A large ohmic drop in the catholyte is expected, which reduces by decreasing the distance to the third electrode and increasing the LiOH concentration in the catholyte, both of which reduce the liquid electrolyte resistance.

It is clear that the effect of the second power supply results mainly in ohmic drop in the electrolyte (Joule effect) and some increase in the Ni electrolyte/liquid electrolyte interface which drives the hydrogen evolution reaction (H_{er}) at the Ni third electrode.

It is common practice in electrochemistry to measure the electrode/electrolyte potential with a reference electrode, i.e. Ag/AgCl reference electrode to separate the effects of ohmic drop and polarization. The authors should do this in order to understand why increasing the voltage of the second potential source, they find a larger lithium collection rate.

Further comments, which should be addressed by the authors

1. Introduction: ion pumping methods based in lithium ion intercalation in battery cathode materials are not referred, they should. See for instance:

Electrochemical methods for sustainable recovery of lithium from natural brines and battery recycling, *Current Opinion in Electrochemistry*, 15, (2019), 102-108.

Direct Lithium Recovery from Aqueous Electrolytes with Electrochemical Ion Pumping and Lithium Intercalation, *ACS Omega* 2021, 6, 51, 35213-35220

Recent advances in reactor design and control for lithium recovery by means of electrochemical ion pumping, *Current Opinion in Electrochemistry*, 35, (2022), 101089.

2. Reaction 3 is limited by the Exchange of lithium ions at the liquid electrolyte/solid LLTO interface. These reactions are slow due to the high hydration energy of $Li(OH)_2^+$ in the loss and recapture of hydration water molecules by the Li^+ ion.

Subsequent diffusion and migration of non hydrated lithium ions takes place at the LLTO Li ion conducting membrane. Diffusion is driven by a concentration gradient and electro-migration by the

electric field, i.e. voltage drop across the LLTO solid membrane.

3. At the Pt/liquid electrolyte and Ni/liquid electrolyte interfaces the Faradaic reactions take place:

These reactions are driven by the potential drop at the Pt(or Ni)/electrolyte interface and follow a Butler-Volmer exponential dependence. The Exchange current density of the HER on Pt is much larger than the OER, so hydrogen evolution is not a slow reaction.

The flux of lithium across the LLTO membrane should equal the flux of O₂ or HO⁻ at the anode because of mass balance of lithium ions and charge balance in the electrolyte.

4. The SHE refers to activity of proton 1, and in alkaline solutions the hydrogen electrode has a lower value. The authors should refer to the REVERSIBLE HYDROGEN ELECTRODE RHE instead of SHE in page 6, lines 236-239:

“In many cases, the potential of the cathode-side surface of LLTO does not drop
137 below the hydrogen-gas-generating potential (0 V vs. standard hydrogen electrode
138 [SHE]) and is always higher than the tetravalent-to-trivalent reduction potential of Ti
139 (0.0 V vs. SHE).”

Lines 143-144

by significantly increasing the overpotential of the oxygen-gas-generating reaction at the anode

should read “decreasing the overpotential”

5. Increasing the secondary-power-supply voltage in the new electrochemical pumping system results in a larger current with a larger ohmic drop and hydrogen evolution.

6. The electrode geometric areas of the Pt and Ni electrodes in the cathode compartment should be defined, as well as the Pt electrode area in the anode compartment, since the total current will be defined by the specific local current given by the Butler-Volmer eqn. and the electrode area.

7. It is not clear if the lithium collection rate comes from chemical analysis of the catholyte or from current I₁. The authors describe “The Li-ion concentration of the cathode-side solution after 1 h of electrochemical pumping was determined by ICP-AES (SPECTROBLUE® FMX26, HITACHI, Japan” But in Fig. the lithium collection rate is plotted.

8. By using reference electrodes at the anode and cathode the authors could separate the effects of interfacial electrode potential acting on the Faradaic reactions, and the electric field driving the lithium ion flux across the LLTO membrane, and the ohmic drops in the electrolytes, particularly the large ohmic drop at the diluted LiOH in the catholyte.

9. The authors should explain clearly the mechanism that leads to the improvement in the new experimental design.

Reviewer #3 (Remarks to the Author):

In this work, the authors designed a novel electrochemical pumping system using three electrodes and two power supplies. And, they demonstrated the performance of the newly designed pumping system in lithium extraction. But, the weakness of this work is the lack of experiments and insufficient discussion supporting the author's claims. Thus, I cannot agree to the publication of this current form of the manuscript. Before reconsideration of the publication, the manuscript should be improved with a major revision.

Specific comments:

1. In P3L68, the authors claimed the electrochemical pumping system exhibits a high selectivity for lithium ions compared to sodium and potassium ions. But, the selective extraction mechanism is ambiguous. Please explain it in detail.
2. Although the authors addressed the electrochemical pumping system as an economic process, it remains unclear. Please carry out the techno-economic analysis of the pumping system compared to other technologies such as adsorption and ion exchange. In particular, recently, the battery system (ACS Omega 2021, 6, 51, 35213–35220; Processes 2022, 10(12), 2654) has been widely examined the lithium-ion recovery. The battery system would recover the consumed energy during extraction, and thus the battery system is considered a very energy-efficient process.
3. In this regard, the proposed pumping system seems to be more complicated than the battery system. Please clarify the pros and cons of the pumping system compared to the battery system.
4. In this work, the information on the lithium source (anode side solution) was not well addressed. Please clarify and justify the rationale for why the authors selected the composition of the solution on the anode side. Is there any target application?
5. The selectivity in lithium extraction could be a pivotal factor governing the system's performance. But, in this work, the selectivity was not well examined. Please show the selectivity results and discuss further them.

Reviewer #1 (Remarks to the Author):

The article titled: "A three-electrode dual-power-supply electrochemical pumping system for ultrafast high-energy-efficiency lithium extraction and recovery" describes the development of a new electrochemical cell set-up allowing faster and improved lithium recovery. The design of the cell is innovative and the research field of great interest; however, additional experiments and clarifications are required before its publication in *Communications Engineering*.

Response: We would like to thank the reviewers and editors for their constructive comments to improve our manuscript. We have carefully considered the reviewers' comments and provided responses to all the points raised by the reviewers. We hope that the new manuscript will meet the standards of *Nature Communications*.

The point-by-point responses to the reviewers' comments are given below.

(Q#1-1) The authors claim that "the Li collection rate of this new system can be limitlessly increased." I doubt the LLTO solid electrolyte will allow an unlimited rate. The lithium diffusion coefficient in its structure is limited. The lithium must migrate through its structure to reach the cathode side, which will roll the rate. What is the value of the lithium diffusion coefficient in LLTO, and how would it affect the "limitlessly" rate proposed?

Response to Q#1-1: Thank you for this insightful comment. As indicated by Fick's law (see below), the diffusion constant, D , is the proportionality coefficient between the Li collection rate (flow rate, J) and the potential gradient (dc/dx), and increases monotonically with increasing potential difference, which is the electrochemical potential gradient.

$$J = -D \frac{dc}{dx}$$

Prof. Inaguma has reported that the diffusion constant of LLTO is estimated to be 3×10^{-8} cm²/s from the Nernst-Einstein equation, assuming that the interaction between diffusing Li ions is small based on electrical conductivity [R1]. As long as the crystal structure of LLTO is maintained, it is theoretically possible to increase the recovery rate indefinitely.

[R1] Y. Inaguma, A Review of Recent Research on Perovskite-Type Lithium Ion-Conducting Oxides. *J. Cryst. Soc. Japan* **58** 62–72 (2016). 10.5940/jcrsj.58.62.

(Q#1-2) The authors also mentioned an "ultrafast high-energy-efficiency." However, I have some doubts about the suitability of such terminology. Increasing the recovery rate by increasing the

voltage applied must have a penalty in the energy consumption required to extract the lithium. There is no calculation regarding the energy needed to extract the lithium in the manuscript; such a value will have a tremendous impact on the potential applicability of this new set-up. These calculations must be included, and the energy consumption per mol of lithium must be included.

Response to Q#1-2: As reviewer #2 pointed out, the energy efficiency required for Li decreases upon increasing the collection rate by increasing the applied voltage. This relationship between recovery rate and energy efficiency is a tradeoff. Figure R1 shows the relationship between the collection rate and energy efficiency depending on the main power-supply voltage when the secondary power-supply voltage is fixed at 10 V. The dependence of energy efficiency on the collection rate in the conventional method, which is the system with one power supply and two electrodes, is also shown. In the conventional method, the energy efficiency significantly decreases upon increasing the Li collection rate due to the occurrence of electronic conduction. In contrast, the new system shows an improvement in collection rate and also an enhancement in energy efficiency upon increasing the applied voltage of the main power source. Therefore, the installation of a secondary power source not only increases the collection rate but also improves energy efficiency.

The orange line in Fig. R1 corresponds to the equation relating the reaction rate to energy efficiency estimated as follows.

In an electrochemical pumping system, the Li recovery amount c_{Li} from the anode side to the cathode side is

$$c_{Li} = \frac{\int I dt}{F} \quad (1)$$

where F and t are the Faraday constant and operating time, respectively. Furthermore, the energy consumption ΔW of the recovery unit during electrochemical pumping is

$$\Delta W = V_{app} \int I dt \quad (2)$$

where V_{app} is the applied voltage. From equations (1) and (2), the energy efficiency E_{ef} can be obtained as follows

$$E_{ef} = \frac{c_{Li}}{\Delta W} = \frac{1}{FV_{app}} \quad (3).$$

The Li collection rate v_{Li} (g/m²/h) is calculated using equation (4), assuming equilibrium among the solution, electrolyte, and applied voltage

$$v_{Li} = \frac{M_{Li} \int_0^t I dt}{F \Delta t} = \frac{M_{Li} V_{app}}{FR_{total}} \quad (4).$$

Combining equations (3) and (4), the relationship between the lithium collection rate and energy efficiency can be expressed as follows:

$$E_{ef} = \frac{M_{Li}}{F^2 R_{total}} \frac{1}{v_{Li}} \quad (5)$$

Figure R1 shows a log-log plot; the orange line has the form

$$\log(E_{ef}) = -A \log(v_{Li}) + B \quad (6).$$

where A and B are fitting parameters. The energy efficiency of the new system can be seen approaching the orange line as the main power-supply voltage increases. This means that we have developed a Li recovery system that achieves high energy efficiency by suppressing electronic conduction due to the reduction of LLTO.

Fig R1. Relationship between collection rate and energy efficiency of Li recovery of the conventional (two electrodes and one power supply) and our method.

Q#1-3) The manuscript shows the improvement obtained with this new set-up; however, the electrolyte used was pure lithium electrolyte. In real samples, lithium concentration is significantly smaller than other cations present in solution (Na, K, Mg...). The increment in the lithium rate could also affect the LLTO capacity to discriminate between these cations; therefore, the method's selectivity in the conditions described must be included, showing the concentration on the cathode-side of the different cations. If other co-cations are co-transported with the increment of the rate, the capacity of this methodology to extract lithium will be compromised; therefore, such measurements are critical for correctly analyzing the proposed method.

Response to Q#1-3: As discussed in the Introduction, the high Li ion selectivity of the LLTO electrolyte for the electrochemical pumping system in the presence of other cationic alkali metals (Na and K) has already been reported in previous studies. Morita et al. [R2] and

Kunugi et al. [R3] have reported that when Li was recovered from solutions with the same concentrations of Li, Na, and K, only Li was recovered; i.e., no Na or K was detected by ICP from the cathode-side solution. The reason why only Li diffuses in LLTO is the crystal structure of LLTO (Fig. R2). LLTO has a perovskite-type structure; when Li is partially substituted at the La site, the La-rich and La-poor layers are stacked in the *c*-axis direction, and Li ions are substituted at the La sites of the La-poor layer. For a Li ion to migrate to a neighboring La site, it must pass through a plane surrounded by oxide ions, which is the bottleneck position for ion diffusion in LLTO. The size of this LLTO bottleneck is 1.12 Å, which is larger than that of the Li ion (ionic radius: 0.92 Å) and smaller than those of the Na ion (1.39 Å) and K (1.64 Å) ions. Therefore, the electrochemical pumping system can concentrate only Li ions from the anode tank to the cathode tank. A description of the crystal structure of LLTO has been added to P. 3, L. 70–P. 4, L. 77 of the revised manuscript.

Fig R2. Crystal structure of $\text{La}_{2/3-x}\text{Li}_{3x}\text{TiO}_3$ ($x = 0.05$) [R4].

[R2] K. Morita et al., *Desalination*, 543, (2022) 116117.

[R3] S. Kunugi et al., *Solid State Ionics* **122** 35–39 (1999).

[R4] Y. Inaguma et al., *J. Solid State Chem.* **166** 67 (2002).

(Q#1-4) On the other hand, increasing the rate will also increase the accumulation of cations on the LLTO surface since these co-cations will be blocked because of LLTO lithium selectivity. How will such charge accumulation affect the recovery process?

Response to Q#1-4: The incorporation of Li ions from the solution into the LLTO solid electrolyte near the anode is not due to electrical potential difference but to chemical

potential differences. In our system, the anode is the positive electrode; Na and K ions do not approach the anode electrode because of Coulombic repulsion, and thus, a blocking layer of cations is not expected to form.

(Q#1-5) How much is the final lithium concentration reached after applying the electrochemical method?

Response to Q#1-5: By continuously applying voltages, it is possible to concentrate LiOH to its saturation concentration, and we have succeeded in precipitating LiOH·H₂O. Depending on the nature of the solution on the cathode side, it is possible to deposit various lithium salts, and we are planning to report these results in the near future.

(Q#1-6) Finally, there are numerous articles directly related to the manuscript topic that should be included in the manuscript: e., g. Recent reviews about the electrochemical ion pumping technology: <https://doi.org/10.1016/j.coelec.2021.100778>; DOI: 10.1002/adma.201905440; or alternative electrochemical cells for lithium recovery: <https://doi.org/10.1016/j.desal.2019.114192>

Values like the selectivity, purity, energy efficiency, energy consumption, and final Lithium concentration in the present work must be compared with the suggested bibliography to analyze the proposed methodology's advantages properly.

Response to Q#1-6: The Li ion selectivity and purity of the recovered solution have been reported in previous studies on the electrochemical pumping method using LLTO (e.g., Morita et al. [R2] and Kunugi et al. [R3]) and are higher than those of other methods. The mechanism of the high ion selectivity is explained in the answer to Q#1-4.

Other advantages of the electrochemical pumping method include that this system can continuously concentrate Li without changing adsorbents/solutions or switching between charging and discharging cycles. Previous reports of Li recovery technologies for batch processing do not include the energy consumed in such process treatments. Comparison of energy efficiency between electrochemical pumping and other systems is difficult. These explanations were added in P. 4, L. 93–96 of the revised manuscript.

As explained in the response to Q#1-5, the final Li concentration in the current study can be concentrated to the saturation concentration of LiOH·H₂O and even precipitated.

Reviewer #2 (Remarks to the Author):

This reviewer finds the present manuscript not suitable for publication for the reasons explained below. Major revision of the manuscript is required before publication.

The authors present a new experimental set up with respect to their previous publication in which an LLTO solid lithium conducting membrane with two platinum electrodes deposited in both faces separates two LiOH solutions of different concentration, on the anode side 1 M LiOH and on the cathode side 0.001 M LiOH solution, with an applied voltage 2.0 V across the LLTO membrane.

In the new set up presented in the manuscript, a third Ni cathode and a second power supply are introduced. Upon applying a second voltage with the second power supply, the real impedance (R_{ct}) of the LLTO decreases and the lithium collection rate increases.

According to data in Figure 1 anode-side solution, 1.0 mol/L aqueous LiOH solution; cathode-side solution, 1×10^{-3} mol/L aqueous LiOH solution; distance between the second and third electrodes, 57 mm. A large ohmic drop in the catholyte is expected, which reduces by decreasing the distance to the third electrode and increasing the LiOH concentration in the catholyte, both of which reduce the liquid electrolyte resistance.

It is clear that the effect of the second power supply results mainly in ohmic drop in the electrolyte (Joule effect) and some increase in the Ni electrolyte/liquid electrolyte interface which drives the hydrogen evolution reaction (HER) at the Ni third electrode.

(Q#2) It is common practice in electrochemistry to measure the electrode/electrolyte potential with a reference electrode, i.e. Ag/AgCl reference electrode to separate the effects of ohmic drop and polarization. The authors should do this to understand why increasing the voltage of the second potential source, they find a larger lithium collection rate.

Response to Q#2: Previous results have shown that the conventional method has high resistance and overvoltage of the reaction of the cathode electrode, which is the hydrogen gas-generation electrode and rate-limiting for the electrochemical pumping reaction [R5]. As explained in the Manuscript, the rate of reaction of this rate-limiting process was enhanced by the application of a voltage by a second power supply at two points: 1) the attraction of Li ions to the third electrode by Coulomb forces decreases the chemical potential near the cathode electrode (2nd electrode) and increases the elution rate from the electrolyte to the solution in the cathode-side tank; 2) the larger surface area of the third electrode also increases the current as the rate of hydrogen gas generation during electrochemical pumping. Impedance measurement results revealed the mechanism of the improved lithium recovery rate. Based on the results so far, we consider that the cause of

the increased Li collection rate is clear, even without the elucidation of the potential profile.

[R5] K. Sasaki et al., *Fusion Eng. Des.* **170** 112500 (2021).

Further comments, which should be addressed by the authors.

(Q#2-1). Introduction: ion pumping methods based in lithium ion intercalation in battery cathode materials are not referred, they should. See for instance:

• Electrochemical methods for sustainable recovery of lithium from natural brines and battery recycling, *Current Opinion in Electrochemistry*, 15, (2019), 102-108.

• Direct Lithium Recovery from Aqueous Electrolytes with Electrochemical Ion Pumping and Lithium Intercalation, *ACS Omega* 2021, 6, 51, 35213-35220

• Recent advances in reactor design and control for lithium recovery by means of electrochemical ion pumping, *Current Opinion in Electrochemistry*, 35, (2022), 101089.

Response to Q#2-1: Thank you for pointing this out to us. We have added the references that you have suggested as Refs. 20–22 in the revised manuscript.

(Q#2-2). Reaction 3 is limited by the Exchange of lithium ions at the liquid electrolyte/solid LLTO interface. These reactions are slow due to the high hydration energy of $\text{Li}(\text{OH}_2)_4^+$ in the loss and recapture of hydration water molecules by the Li^+ ion.

Subsequent diffusion and migration of non hydrated lithium ions takes place at the LLTO Li ion conducting membrane. Diffusion is driven by a concentration gradient and electro-migration by the electric field, i.e. voltage drop across the LLTO solid membrane.

ResponseQ#2-2: Thank you very much for your helpful remarks. As stated in Q#2, in the conventional method (one power source and two electrodes method), the cathode electrode reaction is the rate-limiting step of the electrochemical pumping reaction. Our dual power supply system comprising three-electrodes has solved this problem. Figure R3 shows the concentration dependence of the pH of a LiOH solution. Also shown is the LiOH concentration dependence of the degree of ionization estimated from the pH. This result indicates that at 0.001 mol/L, the ionized lithium ratio is 60%–70%. In contrast, the degree of ionization in a 1 mol/L LiOH solution is only approximately 0.1%. These results show that the concentration of hydrated ions increases as the activity of Li ions increases

with increasing LiOH concentration. In a dual power supply system comprising three electrodes, as proposed in this manuscript, the Li concentration near the second electrode decreases due to the application of voltage to the secondary power supply, as per the answer to Q#2. This effect has been found to allow lithium precipitated from the electrolyte to be eluted into solution without the reaction of recapture to hydrated ions. The hydration reaction of Li ions is considered to occur in the solution of the cathode tank by approaching the third electrode by Coulomb force after the Li ion is eluted from the cathode electrode into solution and is not considered to affect the reaction rate of the cathode electrode.

Fig. R3. LiOH concentration dependency of measured pH of LiOH solution and degree of ionization estimated from pH.

(Q#2-3). At the Pt/liquid electrolyte and Ni/liquid electrolyte interfaces the Faradaic reactions take place:

These reactions are driven by the potential drop at the Pt(or Ni)/electrolyte interface and follow a Butler-Volmer exponential dependence. The Exchange current density of the HER on Pt is much larger than the OER, so hydrogen evolution is not a slow reaction.

The flux of lithium across the LLTO membrane should equal the flux of O₂ or HO⁻ at the anode because of mass balance of lithium ions and charge balance in the electrolyte.

Response to Q#2-3: A LiOH solution with strong basicity was used in this study. Therefore, the solution in the cathode-side tank has a very low concentration of protons and oxonium ions and hydrogen gas generation is slow.

(Q#2-4). The SHE refers to activity of proton 1, and in alkaline solutions the hydrogen electrode has a lower value. The authors should refer to the REVERSIBLE HYDROGEN ELECTRODE RHE instead of SHE in page 6, lines 236-239:

“In many cases, the potential of the cathode-side surface of LLTO does not drop below the hydrogen-gas-generating potential (0 V vs. standard hydrogen electrode [SHE]) and is always higher than the tetravalent-to-trivalent reduction potential of Ti (0.0 V vs. SHE).”

Lines 143-144 “by significantly increasing the overpotential of the oxygen-gas-generating reaction at the anode” should read “decreasing the overpotential”

Response to Q#2-4: Thank you for pointing this out to us. We corrected “SHE” to “RHE”. In Lines 143–144 in the previous version of the manuscript, we thought it proper to use “increase of the overpotential of the oxygen-gas-generating reaction at the anode”. However, the second (Pt) electrode decreases the overpotential of the oxygen gas generation reaction, as Reviewer #2 has pointed out. We changed the sentence in the revised manuscript to clarify this.

(Q#2-5). Increasing the secondary-power-supply voltage in the new electrochemical pumping system results in a larger current with a larger ohmic drop and hydrogen evolution.

Response to Q#2-5: As Reviewer #2 commented, the increase of only the applied voltage of the secondary power supply decreases the total energy efficiency due to the increase of energy of water electrolysis, which is not involved in the lithium recovery reaction in the solution of the cathode-side tank. However, it was found that energy efficiency is increased by increasing the main power-supply voltage. When the applied voltage of the secondary power supply was maintained constant at 10 V, the energy efficiency increased from 0.20 to 1.44 mmol/Wh when the applied voltage of the main power supply was increased from 2 to 5 V. In contrast, the collection rate also increased dramatically from 0.12 to 1.40 kg/m²/h, as shown in Fig. 3a. These

explanations have been added to P. 11, L. 252–256 of the revised manuscript.

(Q#2-6). The electrode geometric areas of the Pt and Ni electrodes in the cathode compartment should be defined, as well as the Pt electrode area in the anode compartment, since the total current will be defined by the specific local current given by the Butler-Volmer eqn. and the electrode area.

Response to Q#2-6: The size of the third electrode ($2 \times 2 \text{ cm}^2$) has already been described in the experimental method section. However, in this manuscript, each resistance, R_i ($i: 1, 2, 3$, LLTO, LiOH) is defined in an equivalent circuit (Fig. 1b), which we regard as including the electrode area ratio.

(Q#2-7). It is not clear if the lithium collection rate comes from chemical analysis of the catholyte or from current II. The authors describe “The Li-ion concentration of the cathode-side solution after 1 h of electrochemical pumping was determined by ICP-AES (SPECTROBLUE® FMX26, HITACHI, Japan)” But in Fig. the lithium collection rate is plotted.

Response to Q#2-7: The text of the experimental method section has been modified for clarity. The portion in quotation mark was added to the following sentence in P.16, L.385: The Li ion concentration of the cathode-side solution after 1 h of electrochemical pumping was determined by ICP-AES (SPECTROBLUE® FMX26, HITACHI, Japan, or Optima 7000DV, Perkin Elmer, USA), and the molar mass of transferred Li, n , was estimated “from the Li -ion concentration of the cathode-side solution.”

(Q#2-8). By using reference electrodes at the anode and cathode the authors could separate the effects of interfacial electrode potential acting on the Faradaic reactions, and the electric field driving the lithium ion flux across the LLTO membrane, and the ohmic drops in the electrolytes, particularly the large ohmic drop at the diluted LiOH in the catholyte.

Response to Q#2-8: In this manuscript, instead of using reference electrodes to determine the overvoltage and voltage drop, the AC impedance method was used to measure the resistance directly to understand the schematic of the new system and clarify the principles of the system. We think that it is better to explain the improvement of the new system over

the conventional method by discussing the results of impedance measurements. The potential profiles suggested by Reviewer #2 are currently under investigation and will be published as a separate paper. Figure R4 shows the potential profile when external electrodes are placed on both sides, although the system is slightly different than that reported herein.

Fig R4. Potential profile of an electrochemical pumping system with 0.1 and 0.001 mol/L LiOH of both side tanks and an applied voltage of 2.0 V.

(Q#2-9) The authors should explain clearly the mechanism that leads to the improvement in the new experimental design.

Response to Q#2-9: Thank you for your comments. The mechanism of improvement of the new system compared to the conventional system is described in P. 5 L. 109 – P. 7 L. 148 of the Introduction.

Reviewer #3 (Remarks to the Author):

In this work, the authors designed a novel electrochemical pumping system using three electrodes and two power supplies. And, they demonstrated the performance of the newly designed pumping system in lithium extraction. But, the weakness of this work is the lack of experiments and insufficient discussion supporting the author's claims. Thus, I cannot agree to the publication of this current form of the manuscript. Before reconsideration of the publication, the manuscript should be improved with a major revision.

Specific comments:

(Q#3-1). In P3L68, the authors claimed the electrochemical pumping system exhibits a high selectivity for lithium ions compared to sodium and potassium ions. But, the selective extraction mechanism is ambiguous. Please explain it in detail.

Response to Q#3-1: The reason why only Li diffuses in LLTO is caused by the crystal structure of LLTO. LLTO has a perovskite-type structure, as shown in Fig. R2 (repeated below); when Li is partially substituted at the La site, the La-rich and La-poor layers are stacked in the *c*-axis direction, and Li ions are substituted at the La sites of the La-poor layer. For a Li ion to migrate to a neighboring La site, it must pass through a plane surrounded by oxide ions, which is the bottleneck position for Li ion diffusion of LLTO. The size of this LLTO bottleneck is 1.12 Å, which is larger than that of the Li ion (ionic radius: 0.92 Å) and smaller than those of the Na (1.39 Å) and K (1.64 Å) ions. Therefore, the electrochemical pumping system can be concentrated only Li ions from the anode tank to the cathode tank. A description of the crystal structure of LLTO has been added to P. 3, L. 70–P. 4, L. 77 of the revised manuscript.

Fig R2 (repeat). Crystal structure of $\text{La}_{2/3-x}\text{Li}_{3x}\text{TiO}_3$ ($x = 0.05$)*

(Q#3-2). Although the authors addressed the electrochemical pumping system as an economic process, it remains unclear. Please carry out the techno-economic analysis of the pumping system compared to other technologies such as adsorption and ion exchange. In particular, recently, the battery system (ACS Omega 2021, 6, 51, 35213–35220; Processes 2022, 10(12), 2654) has been widely examined the lithium-ion recovery. The battery system would recover the consumed energy during extraction, and thus the battery system is considered a very energy-efficient process.

Response to Q#3-2: When comparing economic processes, it is necessary to discuss other factors in addition to energy consumption during the recovery process. In particular, this study uses a continuous process, which is different from batch processing methods for battery systems or adsorption systems. Furthermore, because of the high Li selectivity and purity, there is no need for further purification. Considering these points, it can be estimated that this method has better economic efficiency than other methods.

(Q#3-3). In this regard, the proposed pumping system seems to be more complicated than the battery system. Please clarify the pros and cons of the pumping system compared to the battery system.

Response to Q#3-3: The advantage of the electrochemical pumping system is that it can recover Li ions continuously without the replacement of absorbent materials/solutions or switching between charging and discharging cycles as in other lithium recovery methods. Furthermore, from reaction equations (1) and (2), highly pure hydrogen and oxygen gases are generated from each electrode. These gases can also be recovered and used as fuel. This explanation was added on p. 4 L. 93–p. 4 L. 99 of the revised manuscript.

(Q#3-4). In this work, the information on the lithium source (anode side solution) was not well addressed. Please clarify and justify the rationale for why the authors selected the composition of the solution on the anode side. Is there any target application?

Response Q#3-4: As this paper focusses on a lithium recovery device with a new structure, it was decided that a lithium solution without other cations is better suited. It has already

been shown in previous reports that Li ion selectivity is high even when an anode side solution containing a mixture of other alkali metals is added, as explained in the answer to Q#3-1.

(Q#3-5). The selectivity in lithium extraction could be a pivotal factor governing the system's performance. But, in this work, the selectivity was not well examined. Please show the selectivity results and discuss further them.

Response to Q#3-5: As explained in the Introduction, the high Li ion selectivity of the LLTO ion electrochemical pumping method in the presence of other cationic alkali metals (Na and K) has already been reported in previous studies. For example, Morita et al. [R2] and Kunugi et al. [R3] have reported that when lithium was recovered from solutions with the same concentrations of Li, Na, and K, only Li was recovered; i.e., no Na or K was detected by ICP from the cathode-side solution. The cause of the high ion selectivity was explained in the response to Q#3-1.

[R2] K. Morita et al., , *Desalination*, 543, (2022) 116117.

[R3] S. Kunugi et al., *Solid State Ionics* **122** 35–39 (1999).

REVIEWERS' COMMENTS:

Reviewer #2 (Remarks to the Author):

I am satisfied with the revision of this manuscript by the authors.

Only a small point: Ref. 42 about previous method was replaced now with a paper by Hashinto not the authors, is that correct?

Therefore I recommend publication of the manuscript in the revised form.

Reviewer #3 (Remarks to the Author):

The concerns raised have been effectively resolved, and I'm in favor of publishing without any additional revisions.

REVIEWERS' COMMENTS:

Reviewer #2 (Remarks to the Author):

I am satisfied with the revision of this manuscript by the authors.

Only a small point: Ref. 42 about previous method was replaced now with a paper by Hashinto not the authors, is that correct?

Therefore I recommend publication of the manuscript in the revised form.

Response: Thank you for your kind comments. We have changed the reference number from 42 to 45 in the section you pointed out.